# Insights into the Molecular Mechanisms and Signaling Pathways of Epithelial to Mesenchymal Transition (EMT) in the Pathophysiology of Endometriosis

**DOI:** 10.3390/ijms26157460

**Published:** 2025-08-01

**Authors:** Hossein Hosseinirad, Jae-Wook Jeong, Breton F. Barrier

**Affiliations:** Department of Obstetrics, Gynecology and Women’s Health, University of Missouri, Columbia, MO 65211, USA; shfcn@missouri.edu

**Keywords:** endometriosis, epithelial-to-mesenchymal transition, fibrosis, TGF-β

## Abstract

Endometriosis is a disease characterized by the presence of endometrial glands and stroma outside of the uterine corpus, often clinically presenting with pain and/or infertility. Ectopic lesions exhibit features characteristic of epithelial-to-mesenchymal transition (EMT), a process in which epithelial cells lose polarity and acquire mesenchymal traits, including migratory and invasive capabilities. During the process of EMT, epithelial traits are downregulated, while mesenchymal traits are acquired, with cells developing migratory ability, increasing proliferation, and resistance to apoptosis. EMT is promoted by exposure to hypoxia and stimulation by transforming growth factor-β (TGF-β), platelet-derived growth factor (PDGF), and estradiol. Signaling pathways that promote EMT are activated in most ectopic lesions and involve transcription factors such as Snail, Slug, ZEB-1/2, and TWIST-1/2. EMT-specific molecules present in the serum of women with endometriosis appear to have diagnostic potential. Strategies targeting EMT in animal models of endometriosis have demonstrated regression of ectopic lesions, opening the door for novel therapeutic approaches. This review summarizes the current understanding of the role of EMT in endometriosis and highlights potential targets for EMT-related diagnosis and therapeutic interventions.

## 1. Introduction

Endometriosis is a common gynecological disease that affects approximately 10% of the female reproductive-age population. It is characterized by endometrial tissue found outside of the uterine corpus, associated with varying degrees of subfertility, pain, inflammation, and/or fibrosis [1]. Endometriosis is often limited to the anatomic pelvis, involving the ovary and the parietal and visceral peritoneum, but some variants can be found in the abdominal cavity. Lesions can be found invading into pelvic and abdominal viscera and along deep fibrovascular planes, such as the rectovaginal septum or incisional scar tissue. Case reports and case series report that endometriotic implants can occur at locations distant from the pelvis, such as the pleura, lymphatic system, brain, and pericardium [2].

The term “endometriosis” encompasses a range of biological behavior. Superficial peritoneal endometriosis (SP) is commonly found in the gravity-dependent portions of the pelvis, including the ovarian fossae and superficial ovary, uterosacral ligaments, and in the posterior cul-de-sac [3]. SP is present in 80% of patients with surgically confirmed endometriosis, sometimes coexisting with more invasive variants such as ovarian endometriomas (OMA) and deeply infiltrating endometriosis (DIE) [4,5]. Isolated SP is symptomatic in only 40% of affected individuals. OMA consists of cystic lesions arising within the ovarian parenchyma that can be associated with pain and extensive local fibrosis. DIE is characterized by lesions deep (more than 5 mm) beneath the peritoneal surface and is the variant most often associated with pain. DIE typically extends along deep fascial planes such as the rectovaginal septum and can invade into the muscularis of the pelvic organs. OMA and DIE are both strongly associated with significant inflammation and pain [3,4,5]. It has long been suggested that the phenotypic differences between SP, OMA, and DIE could reflect distinct pathogenic mechanisms [6,7]. Although the presence of pain symptoms varies depending on type, all types, including minimal or superficial, have been linked to impaired fertility [4,5].

The pathogenesis of endometriosis is incompletely understood, and multiple mechanisms are likely involved. The most intensely evaluated hypotheses include retrograde menstruation, coelomic metaplasia, and stem cell migration. Less often evaluated hypotheses include benign hematogenous or lymphatic metastasis or the trans-differentiation of Müllerian rests [8]. The fact that so many different hypotheses exist testifies to the reality that no one theory yet accounts for the plethora of pathological presentations [8]. In addition to epithelial and stromal cells, endometriotic lesions contain immune, and vascular cells, all of which may respond to various signaling cues within the lesion microenvironment. Many clinical questions remain unanswered. If retrograde menstruation is the source, why do detached endometrial fragments escape anoikis? How do they penetrate the peritoneal mesothelial barrier to establish lesions? How can we account for the wide range of invasive behavior or the extent of fibrosis? Why do lesions appear at distal sites? What causes the progesterone resistance observed in many ectopic lesions, as well as in the eutopic endometrium? How do ectopic lesions become established under conditions of relatively low oxygen tension and high oxidative stress?

Epithelial-to-mesenchymal transition (EMT) is a programmed process by which epithelial cells transition to a migratory mesenchymal phenotype and develop resistance to apoptosis [9,10]. During this process, epithelial cells lose apicobasal polarity due to the dissolution of adherent junctions and gain anteroposterior polarity and migratory and invasive capabilities. Growth factors such as transforming growth factor-β (TGF-β) and platelet-derived growth factor (PDGF), as well as hormonal and cytokine signals, including estrogen and interleukin-1β (IL-1β), are known to promote EMT in endometriosis. These factors can act on multiple cell types, including stromal and immune cells, amplifying the inflammatory and fibrotic responses within the lesion [9]. In many cases, the process is reversible, allowing cells to transition back from a mesenchymal to an epithelial phenotype. For this reason, some prefer to call the process epithelial–mesenchymal plasticity (EMP) rather than EMT [11,12]. Recent studies suggest that EMT may account for several hallmarks of endometriosis, including invasiveness, resistance to apoptosis, associated fibrosis, and immunotropism [13,14]. EMT-related transcription factors, such as Snail (Snai1), Slug (Snai2), and ZEB1/2, are frequently upregulated in ectopic lesions, as are EMT-inducing factors such as TGF-β and IL-1β [13,14]. These findings suggest that EMT plays an important role in the pathophysiology of endometriosis. A deeper understanding of this process may open a door to novel, targeted therapies. This review aims to summarize current evidence regarding the role of EMT in endometriosis and to highlight potential EMT-related targets for diagnostic and therapeutic intervention.

## 2. Review Methods

We conducted a review of all indexed full-text reports on EMT-related processes in endometriosis. Particular attention was given to studies evaluating the role of EMT in the development and progression of endometriosis, the availability and predictive value of EMT-related biomarkers in the serum of women with endometriosis, and the effects of EMT modulators on disease progression in cell lines and in vivo animal models of endometriosis. We prioritized recent publications available through PubMed but also included relevant older literature. The search was refined using specific keywords, including “endometriosis”, combined with “EMT”, “E-selectin”, “N-selectin”, “TGF-β”, “fibrosis”, and other related terms. Reports on near-topic areas, including those describing EMT in adenomyosis and endometrial cancer, were generally excluded. Using bibliographical references from our primary review, additional manuscripts were identified and retrieved.

## 3. From Discovery to Classification: Understanding EMT Subtypes

The process of EMT was first observed by developmental biologists [15,16]. Over half a century ago, Elisabeth Hay described the transformation of epithelial to mesenchymal cells during the differentiation of the primitive streak in chicks. Her team later recognized that under some conditions, fully differentiated epithelial cells can transition to a mesenchymal phenotype [15,16]. These foundational discoveries laid the groundwork for understanding EMT as a dynamic and reversible process.

Following these developmental observations, oncology researchers began to report that EMT is critical for epithelial cancers to develop an invasive phenotype [13,17]. Clinicians have long pondered the similarity between DIE endometriosis and locally invasive malignancy. In 1997, Gaetje et al. reported that endometriotic epithelial cells undergo the loss of tumor suppressor gene E-cadherin, while in 2001, Zeitvogel et al. demonstrated that endometriotic epithelial cells express N-cadherin in a manner similar to that of invasive carcinoma cells [18,19]. These findings supported the notion that EMT might be a key driver of endometriotic lesion invasiveness. Indeed, the first papers explicitly suggesting a role for EMT in endometriosis were published in 2012 [20,21].

Based on context and function, EMT is now classified into three subtypes: developmental EMT (Type 1), fibrotic EMT (Type 2), and cancer-associated EMT (Type 3) [13]. Type 1 EMT occurs during embryogenesis and organ development, allowing the epiblast to transform into mesoderm and neural crest tissue. This EMT subtype is tightly regulated; occurs in a non-inflammatory, non-fibrotic context; and is driven by signaling pathways such as TGF-β, nodal, bone morphogenetic proteins (BMPs), fibroblast growth factors (FGFs), and Wnt [22,23]. Type 2 EMT is activated during tissue repair and regeneration, typically in response to injury or inflammation. It involves the transition of epithelial cells into fibroblasts and myofibroblasts, which contribute to extracellular matrix (ECM) deposition and the development of wound fibrosis. Importantly, this subtype is reversible once inflammation resolves, and tissue oxygenation is restored. It involves signaling pathways such as TGF-β/Smad, IGF1/MAPK/ERK, PI3K/NF-kB, and non-canonical Wnt signaling [12]. Type 3 EMT is associated with cancer progression and metastasis. It enables tumor cells to acquire mesenchymal traits, facilitating invasion, migration, and resistance to apoptosis. Type 3 EMT often involves aberrant activation of developmental pathways such as Wnt/β-catenin, TGF-β, Notch, Hedgehog, and PI3K/Akt/mTOR [11]. Together, these three EMT subtypes provide a framework for understanding the diverse biological roles EMT plays in early development, tissue repair, fibrosis, and cancer biology. It has been proposed that Type 2, and to a lesser extent, Type 3 EMT are likely involved in the pathogenesis of endometriosis in the context of repetitive tissue injury and repair [24].

## 4. Molecular Drivers and Mechanisms of EMT in Endometriosis

EMT in endometriosis can be driven by multiple triggering factors, including trans-forming growth factor-beta (TGF-β), hypoxia, estrogen, platelet-derived growth factor (PDGF), obesity, and gut microbiota. These factors activate intracellular signaling pathways that converge on key transcription factors such as Snail (SNAI1), Slug (SNAI2), ZEB1, ZEB2 (SIP1), Twist, and FOXC2, which coordinate the EMT process. As a result, epithelial cells undergo transcriptional reprogramming that leads to loss of epithelial traits and acquisition of mesenchymal features. This includes downregulation of E-cadherin, upregulation of N-cadherin, loss of cytokeratin, increased vimentin expression, and suppression of claudins. Collectively, these molecular changes weaken cell–cell adhesion, increase migratory capacity, and promote the invasive and fibrotic characteristics of ectopic lesions in endometriosis (Figure 1). In the following sections, we explore each of these factors in greater detail.

### 4.1. Factors That Trigger EMT in Endometriosis

#### 4.1.1. Growth Factors

TGF-β is a primary regulator of EMT in many epithelial cell types, including endometriosis. As the major promoter of EMT, it is differentially and abundantly expressed in the endometrium under hormonal control [25]. TGF-β1 upregulation in the endometriotic tissue, serum, and peritoneal fluid of endometriosis patients may be crucial for the development and/or maintenance of this disease [25]. Activation of TGF-β signaling in endometriotic tissue leads to the upregulation of Snail, which can suppress epithelial markers such as E-cadherin and enhance mesenchymal markers like N-cadherin and vimentin [13,14,26]. TGF-β-mediated EMT promotes fibrotic remodeling, which is a hallmark of, but not exclusive to, DIE [27,28]. Activated platelets, which may be periodically present in high numbers in ectopic lesions, release large amounts of TGF-β1, further promoting EMT and lesion persistence [29,30]. Notably, blocking TGF-β signaling in experimental models has been shown to reverse EMT features, reduce lesion invasiveness, and mitigate fibrosis [31,32]. Although not yet evaluated in endometriotic cells, TGF-β induces apoptosis in many epithelial cell lines during the G2/M cell cycle phase but not during the G1/S phase. During G1/S, TGF-β initiates the cellular signaling mechanisms of EMT, while concurrently inhibiting apoptosis [33]. The dual role of TGF-β in promoting EMT and resisting apoptosis might link endometriotic cell survival to lesion invasiveness.

In addition to TGF-β, PDGF signaling also plays a notable role in the EMT process in endometriosis. PDGF acts as a potent mitogen and chemical adsorbent for fibroblasts and mesenchymal cells and plays a crucial role in many processes such as cellular proliferation, differentiation, wound healing, embryonic development, inflammation, survival, and migration [34]. Surrey et al. demonstrated that PDGF exerted a significant dose-dependent effect on endometrial stromal cell proliferation [35]. Yaghoubi et al. showed that PDGF-related gene expression is higher in the peritoneal fluid mononuclear cells of women with endometriosis than in the control group [36]. Recently, multiple studies have shown that PDGF-D plays a critical role in governing EMT [37]. Although we did not find strong evidence linking PDGF to EMT induction in endometriotic tissues, its established role in cancer and fibrosis suggests that it may play a role in the pathophysiology of endometriosis.

#### 4.1.2. Hormonal Regulation

Estrogen, specifically 17β-estradiol, is another potent driver of EMT in endometriosis. It promotes lesion survival and invasion and contributes to hormone resistance by interfering with progesterone receptor signaling and promoting a pro-inflammatory microenvironment [38]. Estradiol increases ZEB1 expression, which directly represses E-cadherin, resulting in weakened cell–cell adhesion and enhanced cell motility [39]. It also activates the RhoA/ROCK pathway, a key regulator of cytoskeletal rearrangement promoting cell contractility and invasiveness [40]. This pathway facilitates actin stress fiber formation, further enhancing cellular migration and resistance to apoptosis [41]. Estradiol has also been shown to increase the expression of hepatocyte growth factor (HGF), which stimulates both EMT and mesothelial-to-mesenchymal transition (MMT) in the peritoneal cavity [42]. Through this mechanism, estradiol not only promotes fibrotic remodeling and lesion attachment but also facilitates peritoneal alterations that support ectopic lesion establishment. Du et al. found that estradiol promotes EMT in endometriosis by increasing MALAT1 expression and suppressing miR-200 family members, especially miR-200c, forming a reciprocal regulatory loop that facilitates estrogen-dependent EMT [40]. Furthermore, estrogen-driven EMT is closely associated with the development of progesterone resistance, which compromises the effectiveness of hormone-based treatments for endometriosis [43].

Progesterone resistance, a hallmark of endometriotic tissues, is intricately linked to EMT. In ectopic lesions, progesterone receptor (PGR) expression is frequently reduced, weakening the regulatory effects of progesterone on epithelial stability [44,45]. This insensitivity allows EMT to proceed unchecked. Notably, studies have shown that high expression of EMT transcription factors like Snail and Slug coincide with reduced PGR levels, suggesting a mechanistic link between EMT activation and hormone resistance [44]. This relationship may underlie lesion persistence and reduced response to hormonal therapy in affected patients.

#### 4.1.3. Environmental and Metabolic Factors

A key environmental factor contributing to EMT in endometriosis is hypoxia. Hypoxia is a powerful environmental trigger of EMT in endometriosis and is theoretically present in retrograde endometrial tissue that has been separated from its blood supply [46]. In animal models, hypoxia-preconditioned endometriotic implants display enhanced growth, which is attributed to the upregulation of angiogenic and proliferative markers such as vascular endothelial growth factor (VEGF) and hypoxia-inducible factors (HIFs) [46]. HIFs are transcription factors composed of two subunits, HIF-α and HIF-β, that form a functional heterodimer. Although both subunits are continuously transcribed under both normoxic and hypoxic conditions, their stability and transcriptional activity are tightly regulated by oxygen availability [46,47]. Xiong et al. demonstrated that hypoxia and EMT are closely linked in the development of endometriosis by showing that HIF-1α–mediated EMT occurs in endometrial epithelial cells from both healthy human endometrial tissue and the eutopic endometrium from endometriosis patients [48]. Hypoxia also promotes VEGF expression, thereby increasing angiogenesis and blood vessel formation to support lesion growth [49]. It also activates TGF-β1 and Wnt/β-catenin signaling, contributing to fibrosis and smooth muscle metaplasia in endometriotic tissues [50]. Furthermore, hypoxia induces autophagy, a cellular survival mechanism that allows endometriotic cells to withstand oxidative stress while preserving EMT-like features [51]. In parallel, it enhances the CXCL12/CXCR4 signaling axis, further boosting cell migration and invasion, theoretically facilitating the establishment and persistence of ectopic lesions [52].

Obesity is another emerging EMT trigger. Obesity-associated factors such as leptin and chronic inflammation drive EMT in cancer cells, promoting a mesenchymal phenotype associated with metastasis [53,54]. Leptin increases vimentin expression and cell migration, while adipokines generally correlate with EMT [55]. Obesity-modified CD4+ T cells have also been shown to promote EMT in prostate cancer cells [56]. Although direct studies in endometriosis are lacking, similar mechanisms may contribute to EMT in this disease.

Additionally, gut microbiota plays a significant role in metabolism, immunity, and chronic inflammation. Studies link gut microbiota to endometriosis progression via effects on estrogen metabolism and immune modulation [57]. Microbes can induce EMT by binding to mucosal layers and disrupting epithelial cell adhesion via bacterial adhesins interacting with E-cadherin/catenin complexes, altering cell polarity and signaling [58]. Fusobacterium infection has been shown to trigger EMT through the MIR4435-2HG/miR-296-5p/Akt2/SNAI1 pathway [59]. Although direct evidence in endometriosis is lacking, these mechanisms likely play a role.

#### 4.1.4. Lipid Signaling

Sphingolipid signaling appears to promote EMT by augmenting the effect of hormones and growth factors. Sphingosine 1-phosphate (S1P) is a bioactive pleiotropic sphingolipid whose cross-talk with factors such as TGF-β is critical for the development of fibrosis [60]. Synthesized by sphingosine kinase and metabolized by SIP lyase, S1P is significantly elevated in ectopic lesions when compared with levels in the eutopic endometrium [61], and the concentration of bioactive S1P is increased in the peritoneal fluid of patients with OMA with respect to the levels for controls [62]. Recent studies indicate that S1P may promote EMT through receptor-mediated signaling cascades. Protein expression of S1PR3 was significantly increased in epithelial cells in OMA and DIE lesions, and the levels correlated with fibrosis and EMT markers. Knockdown of S1PR3 led to reduced ERM/ERK signaling and decreased EMT/fibrosis protein expression [63]. In another study, S1PR1 and S1PR2 were also upregulated in OMA and DIE epithelium, with S1PR2 expression also elevated in the eutopic endometrium from affected individuals [64].

#### 4.1.5. Genetic and Epigenetic Modulators

In addition to protein-level signaling, genetic and epigenetic regulators have been implicated in modulating EMT. Genetic mutations that promote EMT may contribute to the progression of endometriosis. Experimental models have shown that mutations in PTEN, a tumor suppressor gene, can enhance EMT-related features such as increased invasiveness and resistance to apoptosis [65,66]. Additionally, KRAS mutations are known to activate the MAPK/ERK signaling pathway, thereby promoting the expression of EMT markers and stimulating cellular proliferation [67]. Investigators have demonstrated many somatic mutations in DIE, the most frequent of which involves KRAS (26% of DIE lesions) [68]. Similarly, loss of ARID1A, a chromatin remodeling gene, has been associated with enhanced EMT activity, fibrosis, and development of persistent lesions [69]. Although these mutations are commonly found in endometriotic tissues, evidence of causality is mostly circumstantial.

MicroRNAs (miRNAs) serve as post-transcriptional regulators of EMT-related genes. Studies have confirmed that abnormally expressed miRNA can directly or indirectly regulate EMT in endometrial epithelial cells, reduce the adhesion of epithelial cells, increase the invasive capabilities of cells, and eventually exhibit the stromal cell phenotype [70]. The miR-200 family, including miR-200a, miR-200b, miR-200c, miR-141, and miR-429, plays a central role in suppressing EMT by targeting ZEB1/2, maintaining epithelial identity, and preventing E-cadherin downregulation [71,72]. Liang et al. found that miR-200c suppresses endometrial stromal cell migration by targeting MALAT1, a lncRNA that promotes EMT [73]. Other miRNAs such as miR-199a-5p, miR-34c-5p, and miR-126-5p also suppress EMT by targeting ZEB1, Notch signaling, and BCAR3, respectively [70,74,75].

Recent studies have also highlighted the role of circular RNAs (circRNAs) in regulating EMT within the context of endometriosis. For example, circ_0004712 has been shown to promote EMT in endometrial epithelial cells derived from endometriosis patients by binding miR-148a-3p, which leads to WNT10B upregulation and activation of the β-catenin signaling pathway [76]. In contrast, circATRNL1 was reported to inhibit EMT in primary human endometrial epithelial cells from women with endometriosis by sequestering miR-103a-3p, thereby downregulating HMGA2 and reducing mesenchymal marker expression [77]. Additionally, circZFPM2 facilitates EMT-like changes in ectopic endometrial stromal cells by acting as a sponge for miR-122-5p, which leads to KLF4 upregulation, a transcription factor known to promote EMT and fibrosis in endometriosis.

Overall, the induction of EMT in endometriosis arises from the complex interplay of growth factors, hormones, environmental stressors, and lipids, as well as genetic and epigenetic regulators. Key pathways involving TGF-β, hypoxia, estrogen, and PDGF converge to activate EMT transcription factors and signaling cascades, promoting cellular plasticity, fibrosis, and lesion persistence. Moreover, genetic mutations and epigenetic modifications further amplify these processes.

### 4.2. EMT-Related Transcription Factors in Endometriosis

Several transcription factors are known to promote EMT. Among them, Snail (Snai1), a zinc-finger protein, and its closely related counterpart, Slug (Snai2), were the first two key transcription factors discovered [78,79]. Additionally, Twist, a member of the basic helix–loop–helix (bHLH) family, plays a crucial role in initiating the EMT process. Unlike Twist, Snail and Slug repress E-cadherin expression by directly binding to E-box sites in its promoter [80]. More recently, zinc-finger E-box-binding homeobox 1 (ZEB1) has been identified as a major repressor of E-cadherin and other epithelial markers, particularly in EMT, by binding to two high-affinity sites in the E-cadherin promoter region [24]. These transcription factors play a critical role in regulating cellular plasticity and invasiveness in endometriosis.

In a study by Kazmi et al., peripheral blood samples from 75 endometriosis patients across different stages were analyzed and compared with 50 control subjects, revealing a significant upregulation of Snail, Slug, Twist, and ZEB1 in endometriosis cases [78]. Similarly, Bartley et al. observed that Twist and Snail were notably overexpressed in ectopic lesions compared to levels in the normal endometrium [81]. Additionally, Furuya et al. reported that ZEB1 expression was exclusive to epithelial cells in ectopic lesions, while absent in normal endometrium, further suggesting its involvement in endometriosis pathogenesis [82]. A review of the literature indicated that these factors may operate in a hierarchical regulatory network, where Snail and Slug are the initial inducers, activating ZEB family members, Twist, goosecoid, and forkhead box protein C2 (FOXC2) [83]. This interplay likely facilitates disease progression by altering epithelial cell properties.

FOXC2, associated with EMT in renal epithelial cells, is activated by multiple signals, including TGF-β1 and other EMT-inducing transcription factors. Unlike Snail and ZEB1, FOXC2 does not directly repress E-cadherin but instead promotes its cytoplasmic re-localization, influencing cellular adhesion and migration [83]. While direct studies on FOXC2 in endometriosis remain limited, Zhu et al. demonstrated that FOXC2 expression was markedly increased in endometrial carcinoma tissues compared to levels in normal endometrial tissue, suggesting its potential relevance in disease progression [84]. This highlights a possible overlap in EMT-related pathways across various gynecological disorders.

In summary, EMT-related transcription factors such as Snail, Slug, Twist, ZEB1, and FOXC2 are central to the regulation of epithelial plasticity and invasive behavior in endometriosis. Their upregulation in endometriotic tissues underscores their role in repressing epithelial markers, promoting mesenchymal traits, and contributing to progesterone resistance. Understanding the intricate network of these regulators provides valuable insight into the molecular basis of endometriosis and may offer novel therapeutic targets for managing disease recurrence and treatment resistance.

### 4.3. Molecular Alteration in EMT in Endometriosis

The transition from epithelial to mesenchymal phenotype in endometriosis involves a coordinated downregulation of epithelial markers and upregulation of mesenchymal markers. These molecular changes weaken cell–cell adhesion, enhance cellular motility, and promote invasive behavior in ectopic lesions. The following are key molecular alterations associated with EMT in endometriosis.

#### 4.3.1. Cadherin Switch

E-cadherin is a calcium-dependent transmembrane glycoprotein crucial for maintaining epithelial cell–cell adhesion and tissue polarity. In contrast, N-cadherin is predominantly expressed in mesenchymal cells and facilitates cell migration and tissue remodeling [85,86]. During EMT, epithelial cells undergo a cadherin switch characterized by repression of E-cadherin and induction of N-cadherin, which enhances cellular motility and invasive potential [81,87]. Studies by Cai and Bartley demonstrated reduced E-cadherin and increased N-cadherin expression in ectopic endometrial tissues, correlating with elevated levels of EMT transcription factors such as Snail and Twist [88]. Liu et al. further reported that EpCAM overexpression coincides with this cadherin switch, promoting EMT and disease progression [89].

#### 4.3.2. Loss of Cytokeratin

Cytokeratins (CKs) are intermediate filaments maintaining epithelial structure and polarity. Downregulation of CKs, including CK18, is a hallmark of EMT and indicates reduced epithelial characteristics and enhanced cellular motility [90,91]. Matsuzaki et al. found decreased cytokeratin expression in epithelial cells of ectopic lesions compared to levels in the menstrual endometrium [20]. While further studies are needed, this shift supports a mesenchymal transition in endometriosis.

#### 4.3.3. Upregulation of Vimentin

Vimentin is an intermediate filament protein predominantly expressed in mesenchymal cells, where it supports cell structure, cytoskeletal integrity, and resistance to mechanical stress [90,91]. During EMT, vimentin expression is significantly upregulated, contributing to cytoskeletal remodeling and increased cellular motility. Elevated vimentin levels are a well-established marker of EMT and are commonly detected in invasive endometriotic cells [88,92]. Increased vimentin expression has been observed in epithelial cells of peritoneal lesions and deep infiltrating endometriosis, suggesting a role in EMT-associated structural changes [20]. This upregulation correlates with reduced E-cadherin levels and increased expression of TGF-β1 and EMT-related transcription factors, supporting its role in promoting cell motility and invasiveness [88]. Additionally, higher vimentin levels in endometrial stromal cells have been linked to downregulation of miRNA-223, enhancing cell migration, invasion, and proliferation [93]. These findings highlight vimentin as a key contributor to EMT in endometriosis, driving cellular changes that support lesion development and disease progression.

#### 4.3.4. Suppression of Claudins

Claudins are transmembrane proteins forming tight junctions in epithelial cells, regulating paracellular permeability and maintaining cell polarity with barrier-forming claudins (e.g., claudin-1, -3, -5, -11, -14, -18) predominant in tight epithelia [94]. Horne et al. demonstrated the localization of claudin-7 and claudin-11 in epithelial cells of both eutopic and ectopic endometrium, reporting impaired claudin-11 localization in ectopic tissue, suggesting a partial EMT [94]. Similarly, Gaetje et al. found decreased expression of claudin-3, -4, and -7 in peritoneal ectopic lesions compared to levels in human endometrium [95], while Pan et al. also observed downregulated claudin-3 and claudin-4 in ectopic endometrium [96]. Table 1 summarizes EMT triggers, transcription factors, and molecular alterations associated with EMT in endometriosis. These findings indicate that altered localization and reduced expression of claudins may contribute to EMT in endometriosis, potentially facilitating the invasiveness and progression of ectopic endometrial lesions.

## 5. Fibrosis

Fibrosis is primarily defined as the excessive accumulation of extracellular matrix (ECM) components, predominantly collagen, that arises during wound healing and chronic inflammatory processes [97]. It consists of the progressive build-up of connective tissue resulting from sustained inflammation or tissue damage, potentially leading to irreversible organ dysfunction or failure [97].

Ectopic lesions are frequently associated with chronic inflammation and fibrosis, unlike the normal eutopic endometrium, which rarely develops fibrosis despite recurrent cycles of dissolution and bleeding. Fibrosis is a key determinant of disease severity and is believed to contribute to common endometriosis-associated symptoms, including pain and infertility [98]. During chronic inflammation, persistent EMT promotes fibrosis by transforming epithelial cells into mesenchymal-like myofibroblasts capable of secreting ECM [99]. EMT is critical for the development of fibrosis in various tissues under pathological conditions, such as pulmonary fibrosis, liver cirrhosis, and systemic sclerosis. During the progression of endometriosis, EMT facilitates the development of invasive characteristics by ectopic lesions through the transition of epithelial cells to a mesenchymal phenotype, thus enhancing their fibrotic and invasive features [98,99]. Ectopic endometriotic lesions, particularly in rectovaginal endometriosis, often contain smooth muscle cells believed to arise through smooth muscle metaplasia (SMM), a process in which fibroblasts transdifferentiate into myofibroblasts [100].

TGF-β1 is a potent inducer of both EMT and SMM [101]. TGF-β1 signaling activates pathways such as Smad, Wnt/β-catenin, focal adhesion kinase (FAK), and Rho-associated protein kinase (Rho/ROCK), all of which promote fibrotic processes [101,102]. Many studies have confirmed an association between TGF-β and Wnt/β-catenin signaling pathways in endometriotic tissues [24]. Both TGF-β and the Wnt/β-catenin signaling pathway play a key role in promoting endometriosis-associated fibrosis. Dysregulated activation of Wnt3/β-catenin has been implicated in driving fibrotic processes triggered by the presence of persistent endometriotic lesions. Additionally, TGF-β1 signaling enhances fibrosis by stimulating platelet activation, attracting macrophages, and promoting neuropeptide release, all of which amplify fibrotic responses and exacerbate disease progression [102,103]. Blocking the TCF/β-catenin pathway can decrease the expression of myofibroblast markers such as α-SMA, collagen I, and fibronectin in endometriotic stromal cells [104].

Overall, fibrosis is a hallmark of endometriosis that contributes to both the structural complexity of lesions and the severity of clinical symptoms. TGF-β1 signaling plays a pivotal role in driving fibrogenesis by promoting the development of ECM-secreting mesenchymal cells during EMT. TGF-β1 likewise promotes SMM, which enhances populations of myofibroblasts that promote tissue remodeling and fibrosis. Because TGF-β1 and Wnt/β-catenin serve as central regulators of both EMT and SMM/fibrosis in ectopic lesions, these pathways provide a potential therapeutic target.

## 6. Lesional Differences

### 6.1. Lesional Patterns of EMT Expression

The expression of EMT markers varies significantly across different types of endometriotic lesions. Early-stage red peritoneal lesions, considered more active, exhibit decreased levels of epithelial markers such as E-cadherin and cytokeratin and increased expression of the mesenchymal marker vimentin, an expression profile consistent with active EMT. In contrast, more established black peritoneal lesions show co-expression of E-cadherin and vimentin, indicating a partial or less active EMT phenotype [20].

OMA displays a more mesenchymal phenotype, with reduced epithelial markers (E-selectin, cytokeratin) and increased vimentin expression, suggesting a more complete EMT process. DIE lesions, on the other hand, appear to retain some epithelial features, with strong expression of E-selectin and increased dephosphorylated β-catenin—markers more typically associated with an epithelial phenotype [20].

### 6.2. Microenvironmental Influences on EMT Dynamics

Matsuzaki et al. investigated the role of local matrix stiffness and TGF-b1 signaling on EMT induction in two lesion types (superficial red lesions and DIE) and in the eutopic endometrium from patients with DIE. Using polyacrylamide gels of varying stiffness, they demonstrated that stiffer matrices enhanced TGF-β1-induced EMT, resulting in a reduced in cell–cell contacts and decreased expression of epithelial markers such as E-cadherin and ZO-1 [20].

In vivo analysis revealed further differences between lesion types: DIE lesions exhibited high E-cadherin and minimal nuclear p-Smad2/3 staining, indicating low TGF-β1 activity. Conversely, red peritoneal lesions showed increased E-cadherin and increased nuclear p-Smad 2/3 expression, reflecting active TGF-β1 signaling. The authors proposed that EMT is initiated during early lesion implantation due to higher local TGF-β1 levels, but may reverse in DIE lesions over time as TGF-β1 signaling diminishes, allowing epithelial characteristics to re-emerge [105]. However, this view is challenged by findings from Liu et al., who reported robust TGF-β1 activity and elevated markers of EMT, fibroblast-to-myofibroblast transition (FMT), smooth muscle metaplasia (SMM), and fibrosis in both OMA and DIE, with DIE exhibiting even greater EMT-associated change [106].

### 6.3. Molecular Feedback Loops and Lesion-Specific Regulation

A study by Ntzeros et al. compared EMT-related gene expression between OMA and DIE, focusing on the ZEB1/E-cadherin/ miR-200b axis. Although ZEB1 expression did not differ significantly between lesion types, its expression in the eutopic endometrium of OMA patients was lower than from healthy controls—a pattern not observed in patients with DIE. Moreover, E-cadherin levels were markedly lower in OMA cyst walls compared to the levels in matched eutopic tissue, while DIE lesions did not show this reduction. MiR-200b, which suppresses EMT, was elevated in OMA but not in DIE. These findings support lesion-specific differences in the regulation of EMT, with OMA displaying more mesenchymal features and DIE retaining a more epithelial profile [107].

### 6.4. Intra-Lesional Heterogeneity

Further complexity is introduced by the possibility of differential expression of EMT-related proteins within a single lesion. Donnez et al., using a baboon model of DIE, found that mitotic activity (Ki-67 staining) was increased and β-catenin expression decreased at the invasion front compared to at the lesion center. Interestingly, E-cadherin expression remained consistent throughout. These findings suggest a pattern of collective cell migration, rather than individual cell invasion, as well as spatial heterogeneity in EMT marker expression within individual lesions [108]. Although interspecies variation must be considered, these findings emphasize the importance of accounting for potential regional variation within lesions when interpreting EMT dynamics.

In summary, EMT marker expression in endometriosis varies not only between lesion types but may also vary within different regions of the same lesion. Active EMT, marked by reduced E-cadherin and elevated mesenchymal markers, is more prominent in early-stage red peritoneal lesions and OMA, potentially driven by high TGF-β1 activity. In contrast, DIE may retain more epithelial traits, possibly due to reduced TGF-β1 signaling, although this finding was inconsistent. These observations underscore a dynamic EMT–MET balance influenced by the lesion’s local microenvironment, including matrix stiffness and the local signaling context.

## 7. Biomarkers

While laparoscopy remains the current gold standard for the diagnosis of endometriosis, its invasiveness and limited specificity underscore the need for reliable, non-invasive biomarkers. In one study, laparoscopy demonstrated a sensitivity of 90.1% (95% CI: 81.0–95.1), a low specificity of 40.0% (95% CI: 23.4–59.3), and a corresponding positive predictive value of 81.0% and a negative predictive value of 58.8 [109]. Based on these parameters, the single point estimated area under the receiver operating characteristic curve (AUC) was 0.70, representing acceptable but not excellent diagnostic accuracy [109]. Emerging evidence suggests that serum-based markers related to EMT including serum microRNAs (miRNAs), cadherins, and hypoxia-inducible factor 1-alpha (HIF-1α) offer promising, non-invasive alternatives for diagnosis. Several candidate biomarkers have demonstrated moderate to high diagnostic accuracy in early clinical validation studies (Table 2 and Table 3).

### 7.1. EMT-Associated Serum microRNAs

#### 7.1.1. miR-17-5p, miR-20a, and miR-22

A 2013 study identified 27 differentially expressed miRNAs in the serum of endometriosis patients, with miR-17-5p, miR-20a, and miR-22 showing strong downregulation relative to that of the controls [110]. These miRNAs yielded AUCs of 0.74 (95% CI: 0.58–0.90), 0.79 (95% CI: 0.65–0.93), and 0.85 (95% CI: 0.71–0.98), respectively. Sensitivity and specificity were not reported. Although not explored in endometriosis, these miRNAs are known to modulate EMT-related pathways in malignancy. For instance, miR-17-5p directly targets vimentin in colorectal cancer [114], miR-20a inhibits EMT by suppressing TWIST1 and TGFBR2 in breast cancer [115], and miR-22 blocks Snail expression and the MAPK1/Slug/vimentin feedback loop in bladder cancer [116].

#### 7.1.2. miR-200 Family

The miR-200 family (miR-200a, miR-200b, and miR-141) is implicated in EMT suppression through downregulation of ZEB1 and ZEB2 (SIP1) transcriptional repressors of E-cadherin [117]. A 2015 study demonstrated that plasma levels of these miRNAs were significantly reduced in patients with endometriosis and exhibited a diurnal pattern, with higher concentrations observed in evening samples [72]. In an analysis of sera obtained in the evening from 32 patients with endometriosis and 24 controls, the combination of these three miRNAs yielded an AUC of 0.76 (95% CI: 0.63–0.87), with a sensitivity of 84.4% and a specificity of 66.7%. When assessed individually, the AUCs were 0.75 (miR-200a), 0.67 (miR-200b), and 0.71 (miR-141), while the sensitivity and specificity were respectively 90.6% and 62.5%, (miR-200a), 90.6% and 45.8% (miR-200b), and 71.9% and 70.8% (miR-141).

#### 7.1.3. Combined Serum miRNA Panels

A 2013 high-throughput profiling study evaluating 765 serum miRNAs in pooled samples (*n* = 10 per group) identified miR-199a and miR-122 as significantly upregulated in endometriosis, while miR-145, miR-141, miR-542-3p, and miR-9 were significantly downregulated [112]. AUC values for individual miRNAs were as follows: miR-199a, 0.83 (95% CI: 0.73–0.92); miR-122, 0.84 (95% CI: 0.75–0.92); miR-145, 0.88 (95% CI: 0.81–0.95); miR-542-3p, 0.85 (95% CI: 0.77–0.94); miR-141, 0.85 (95% CI: 0.77–0.93); and miR-9, 0.83 (95% CI: 0.74–0.92). The authors developed a diagnostic panel comprised of miR-199a, miR-122, miR-145, and miR-542-3p that achieved an AUC of 0.99 (95% CI: 0.98–1.00), with a sensitivity of 93.2% and a specificity of 96.0% for the detection of endometriosis. Notably, higher serum levels of miR-199a and miR-122 were also reported to correlate with advanced (stage III/IV) versus early-stage (I/II) disease (*p* < 0.05), although detailed subgroup analyses were not reported.

A 2025 study by Chen et al. evaluated a serum panel comprised of miR-141, miR-199a, miR-122, miR-145, and CA-125 in a cohort of 155 infertile patients with laparoscopically confirmed endometriosis and 77 infertile controls [111]. The diagnostic panel achieved an AUC of 0.94 (95% CI: 0.898–0.980), with a sensitivity of 81.8% and a specificity of 92.6%. Individually, miR-145 performed best, with an AUC of 0.76 (95% CI 0.68–0.85), while miR-141 had the lowest discriminatory value, with an AUC of 0.588 (95% CI 0.51–0.67). The authors also reported that a separate panel comprised of miR-199a, miR-122, miR-145, and CA125 was able to distinguish stage I/II from III/IV endometriosis with an AUC of 0.76 (95% CI 0.65–0.87), a 79.6% sensitivity, and a 73.5% specificity.

### 7.2. EMT-Related Serum Protein Markers

Serum concentrations of EMT-associated proteins have also been investigated. In a study of 64 in vitro fertilization (IVF) patients (32 with surgically confirmed endometriosis, 32 controls), levels of E-cadherin, N-cadherin and HIF-1α were measured [118]. Compared to controls, patients with endometriosis had lower serum E-cadherin: 107.6 ± 31.9 vs. 137.9 ± 94.5 ng/5 µg protein (*p* < 0.05); higher N-cadherin: 525.3 ± 214.8 vs. 286 ± 118.6 (*p* < 0.01), and higher HIF-1α: 540 ± 222 vs. 272 ± 84 (*p* < 0.001). The discriminatory performance of these markers was modest: the AUC for E-cadherin was 0.62 (95% CI: 0.33–0.81); for N-cadherin, it was 0.71 (95% CI: 0.45–0.86); and for HIF-1α, it was 0.74 (95% CI: 0.48–0.88). These findings suggest that serum N-cadherin and HIF-1α provide moderate discriminatory value, while E-cadherin is of limited diagnostic value.

Numerous EMT-related serum biomarkers, including individual and clustered miRNAs as well as protein factors have shown the potential to differentiate patients with endometriosis from controls with reasonable accuracy. Diagnostic performance varies across studies, and some variability may be attributable to differences in sample timing (e.g., time of day), which is known to influence circulating miRNA levels [117,119]. Future studies should aim to validate these biomarkers in larger, diverse populations, account for temporal and hormonal variables, and incorporate clinical phenotypes to enhance diagnostic specificity and staging utility.

## 8. Emerging Interventions That Target EMT

Current treatments for endometriosis include both medical and surgical approaches aimed at alleviating symptoms and preserving fertility. Medical therapies primarily involve hormonal treatments such as progestins, combined oral contraceptives, and GnRH analogs, which work by lowering estrogen levels to suppress lesion activity [120]. A deeper understanding of the role of EMT in the pathophysiology of endometriosis has opened new avenues for therapeutic intervention. Several pharmacological candidates, including plant-derived compounds traditionally used for dysmenorrhea, have emerged as potential EMT modulators. These agents have shown promising effects in preclinical models by inhibiting EMT and reducing lesion progression. It is important to note that most of these findings are based on in vitro and animal studies and have not yet been translated into clinical use. In this section, we summarize compounds that suppress EMT and limit lesion development, based on preclinical evidence (Table 4).

### 8.1. Isoflavonoids

Isoliquiritigenin (ISL) is a natural flavonoid from Glycyrrhiza uralensis (licorice root) and Allium cepa (shallot), known for its antioxidant and anti-inflammatory properties [127,128,129]. In a Balb/c mouse endometriosis model, ISL treatment increased E-cadherin expression in lesions, and reduced N-cadherin, Slug, and Snail. It significantly decreased lesion size and weight, reduced inflammatory cytokine levels, and induced apoptosis by upregulating BAX and caspase-3 and downregulating Bcl-2 [121]. Treatment with ISL also reduced the proliferation and migration of human endometriotic End1/E6E7 cells, in which it also inhibited the expression of N-cadherin, Snail, and Slug and increased the expression of E-cadherin.

3,6-dihydroxyflavone (3,6-DHF), found in fruits and vegetables, exhibits anti-cancer and anti-invasive effects [130,131]. In both a SCID mouse human endometriosis model, and a Sprague Dawley rat endometriosis model, 3,6-DHF was found to decrease lesion size in a dose-dependent manner. In primary human OMA stromal cells treated with different concentrations of 3,6-DHF (0, 5, 10, and 20 μM, respectively) for 24 h and then analyzed by Western blot, protein expression of E-cadherin was increased, and N-cadherin, Twist, Snail, and Slug were decreased in a dose-dependent manner. 3,6-DHF also inhibited the growth and migration of cells and downregulated Notch signaling components (Notch1, NICD, Hes-1) [122].

### 8.2. Plant Alkaloids

Tetramethylpyrazine (TMP), derived from Rhizoma chuanxiong, exhibits anti-fibrotic and anti-nociceptive properties. Huang et al. co-cultured isolated human OMA stromal cells or 11z human endometriotic epithelial cells with thrombin-activated platelets and then treated the cultures with two concentrations of TMP: low-dose (25 μMol/mL) and high-dose (100 μMol/mL). TMP inhibited the platelet-induced activation of TGF-β1, downregulated vimentin and fibronectin in a dose-dependent manner, and reversed platelet-suppressed protein expression of E-cadherin in both cell types. In vivo, TMP reduced lesion weight, fibrosis, and hyperalgesia in Balb/c mice, inhibited EMT and fibroblast-to-myofibroblast transition (FMT), and attenuated fibrogenic signaling [123].

### 8.3. Terpenes

Parthenolide, a sesquiterpene lactone from Tanacetum parthenium (feverfew), inhibits cancer cell invasion and proliferation [132,133]. In a Wistar albino rat endometriosis model, parthenolide administered at doses of 2 mg/kg and 4 mg/kg reduced lesion size in a dose-dependent manner. PCR and Western blot analysis of the lesions revealed inhibition of EMT via downregulation of the PI3K/AKT/GSK-3β/β-catenin signaling pathway, with decreased cytoplasmic β-catenin levels. This effect was associated with PTEN and E-cadherin upregulation and vimentin downregulation. Lesions from rats treated with parthenolide exhibited an increase (*p* < 0.05) in the tissue levels of caspase-3, BAX, and the ratio of BAX/ Bcl-2, with Bcl-2 decreased in a dose dependent manner. Human immortalized 12z endometriotic epithelial cells treated with 5 υM parthenolide for 48 h were found to display significantly decreased migratory and invasive properties, as measured by wound healing and Transwell assays [124].

### 8.4. Polysaccharides

Fucoidan (FC), a sulfated polysaccharide from Fucus vesiculosus (brown seaweed), exhibits anti-angiogenic and anti-tumor and activity [134,135]. In a Balb/c mouse model of endometriosis, three concentrations of fucoidan (low dose = 10 mg/kg; mid dose = 50 mg/kg, and high dose = 150 mg/kg) were administered for 42 days by oral gavage. Treatments resulted in a dose-dependent decrease in lesion volume and weight. Western blot analysis of lesions revealed a reduction in BAX and an increase in Bcl-2

In the human endometriosis cell lines End1/E6E7 and Vk2/E6E7, FC counteracted estradiol-induced EMT by restoring E-cadherin levels and reducing the expression of N-cadherin, Slug, Snail, and vimentin. FC treatment also led to decreased cell migration in these cell lines [92].

### 8.5. Biguanides

Metformin, widely used in type 2 diabetes, has demonstrated anti-inflammatory effects via inhibition of NFκB [136]. In a study of primary human OMA stromal cells, the authors found that 72 h of treatment with 5 mM, metformin abrogated the proliferative effects of 17 β-estradiol in culture and inhibited cell migration and invasion. Metformin treatment inhibited expression of Snail, Twist, vimentin, and N-cadherin and decreased the nuclear localization of β-catenin in a dose-dependent manner, as evidenced by Western blot analysis [125]. Nuclear localization of β-catenin in OMA stromal cells can activate the Snail promoter [137]. The authors concluded that the inhibitory effect of metformin on EMT in OMA stromal cells was therefore related to the observed decrease in nuclear β-catenin [125].

### 8.6. S1P Receptor Modulators

Sphingosine 1-phosphate (S1P) is a signaling lipid that promotes EMT and fibrosis through various receptor isoforms. In a Balb/cA mouse model of endometriosis, the non-selective S1P inhibitor FTY720, when administered intraperitoneally at a concentration of 1 mg/kg per day for 14 days, was effective in reducing lesion size, fibrosis, Type I collagen levels, and levels of inflammatory cytokines [126]. However, FTY720, also known as fingolimod, is additionally known to cause clinically significant cardiovascular side effects [138]. S1P is formed from sphingosine by sphingosine kinase-1 (SphK1). A different study found that primary human OMA stromal cells treated with 10 ng/mL TGF-β experienced an 11.5-fold upregulation of SphK1. Investigators conditioned primary OMA stromal cells with various concentrations of S1P and treated half with JTE013, a selective S1PR2 antagonist. Although EMT markers were not evaluated, treatment with high dose (125 nM) S1P for 8 h resulted in a 20% increase in cell number, and this effect was abrogated by co-treatment with JTE013 [139]. Taken together, these two studies suggest that S1P antagonists may have an inhibitory effect both on the proliferation of OMA stromal cells and on their role in the development of fibrosis.

Targeting EMT represents a promising therapeutic strategy for the management of endometriosis. A diverse array of pharmacological compounds, ranging from naturally derived flavonoids and alkaloids to clinically approved agents like metformin, have demonstrated the ability to suppress EMT and attenuate lesion development in both in vitro and in vivo models. Across studies, a consistent pattern emerges: these interventions commonly upregulate epithelial markers such as E-cadherin, while downregulating mesenchymal markers including N-cadherin, Snail, Slug, and vimentin. Additionally, several agents effectively modulate key signaling pathways involved in EMT, such as Notch, PI3K/AKT, β-catenin, and TGF-β/Smad, leading to reduced lesion size, fibrosis, and inflammation. Collectively, these findings support EMT as a viable therapeutic target in endometriosis and highlight several promising candidates for further preclinical and clinical development.

## 9. Conclusions

Cellular processes associated with EMT appear to play a critical role in the development and progression of endometriosis. Activation of transcriptional regulators such as Snail, Slug, Twist, and ZEB1/2 lead to the loss of endometriotic epithelial traits and acquisition of mesenchymal traits, thus enhancing lesion motility, invasion, and resistance to apoptosis. Local hypoxia/oxidative stress factors such as HIF-1α, local immune factors such as TGFβ, and local environmental factors such as matrix density appear to contribute to the development of endometriosis by triggering EMT processes. EMT in endometriosis appears to most closely resemble Type II tissue repair EMT, a process driven by inflammatory factors related to local tissue injury and platelet activation. The plasticity of this process appears to lead to incomplete or partial EMT in ectopic lesions, with a pattern of invasion more reflective of collective tissue migration than metastasis of individual cells. The involvement of multiple environmental signals in the EMT process suggests that the phenotype of each endometriotic lesion may be determined by factors in the local microenvironment. Knowledge of EMT processes in endometriosis has led to the identification of multiple candidate biomarkers for non-invasive diagnostic testing. A limited number of small studies have found that the plasma concentrations of some EMT-related factors, mostly circulating microRNAs, appear to correlate with the presence of endometriosis. Elucidation of the EMT pathways upregulated in endometriosis has aided in the identification of potential novel therapies. In vitro and in vivo animal models have been used to demonstrate the effect of therapeutic modulation of EMT-related pathways on the growth and progression of lesions and degree of fibrogenesis. To date, there have been no published clinical trials of these therapies in women.

## 10. Future Directions

Future studies should help to clarify a few important points. First, the natural evolution of lesional expression of EMT-related molecules is unknown. As previously noted, active EMT processes are found to occur to a greater degree in red lesions than in black lesions. This suggests a temporal change in EMT expressions as lesions mature, but this remains to be proven. Second, it would be helpful to map the distribution of EMT-related molecules in individual cells within a single lesion. Evidence of differential intra-lesional expression has been reported in a baboon model of DIE but has not yet been characterized in human endometriosis. This knowledge would be important for the standardization of sample collection and analysis and might better inform our understanding of the local microenvironment. Third, the mechanism of action of treatments that reverse EMT and alter the natural history of endometriosis in animal models should be more fully characterized in order to inform the selection of therapies that employ synergistic mechanisms of action. Finally, EMT-related serum biomarkers can reportedly detect endometriosis with a high degree of sensitivity and specificity, based on small studies. Larger clinical trials are now needed. There is an opportunity to use EMT-related biomarkers in combination with other known biomarkers, such as CA-125 or IL-6, to improve predictive values and potentially inform the development of methods for functional disease staging.

## Figures and Tables

**Figure 1 ijms-26-07460-f001:**
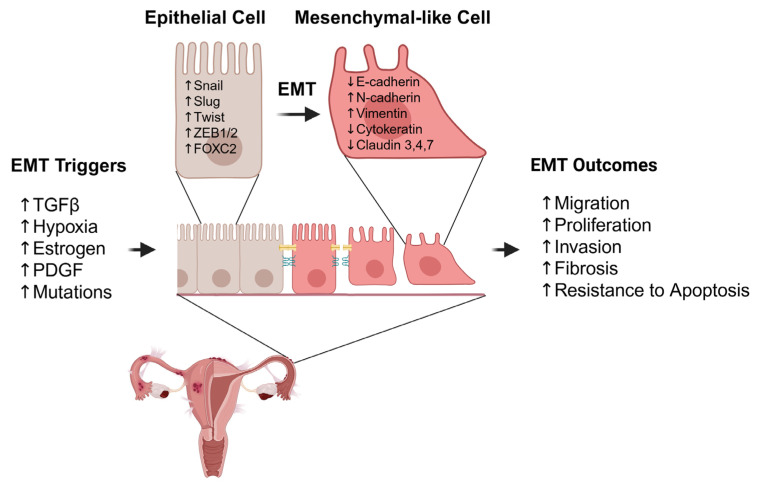
Epithelial-to-mesenchymal transition (EMT) and its role in endometriosis pathophysiology. EMT is triggered by various stimuli, including TGF-β, hypoxia, estrogen, PDGF, and genetic mutations. These triggers induce the expression of EMT transcription factors such as Snail, Slug, Twist, ZEB1/2, and FOXC2, leading to the transition of epithelial cells into mesenchymal-like cells. This transition is marked by decreased epithelial markers (e.g., E-cadherin, cytokeratin, claudin 3/4/7) and increased mesenchymal markers (e.g., N-cadherin, vimentin). As a result, cells gain enhanced migratory capacity, proliferative ability, invasiveness, resistance to apoptosis, and contribute to fibrosis in endometriosis lesion formation and progression.

**Table 1 ijms-26-07460-t001:** EMT-related triggers, transcription factors, and molecules in endometriosis.

Category	Molecule/Factor	Change in Endometriosis	Function in EMT	Reference
Key Triggers	TGF-β	↑ Active in lesions	Master EMT inducer by upregulated EMT-TFs (Snail, ZEB1); involved in fibrosis.	[27,28,29,30]
	PDGF	↑ At lesion sites	Platelet-derived signaling promotes invasion under hypoxia.	[36,37]
	Estradiol	↑ Estrogen-rich environment	Upregulates ZEB1, activates RhoA/ROCK and HGF pathways, and promotes EMTEMT gene regulator under stress by activating VEGF, Wnt/β-catenin, and TGF-β1 pathways.	[40,41,42]
	Hypoxia (HIF-1α/2α)	↑ Elevated in ectopic tissue	EMT gene regulator under stress by activating VEGF, Wnt/β-catenin, and TGF-β1 pathways.	[46,47,50]
	S1P	↑ At lesion sites	Cross-talk with factors such as TGF-β.	[60,62]
	Genetic mutations in PTEN, KRAS, ARID1A	Frequently mutated in DIE	Promote EMT via loss of suppression or gain of oncogenic signals.	[65,66,67,69]
Transcription Factors	Snail (SNAI1), Slug (SNAI2)	↑ Upregulated	Induced by TGF-β, hypoxia, estrogen, and repressed by E-cadherin.	[81]
	ZEB1/ZEB2	↑ Upregulated	Targeted by miR-200 family and promotes EMT.	[24]
	Twist	↑ Upregulated	Associated with initiation and invasion of EMT.	[78]
	FOXC2	Likely ↑	Enhance cell motility.	[83]
Epithelial Markers	E-cadherin	↓ Downregulated	Repressed by Snail, Slug, ZEB1/2 and weakening of cell–cell adhesion.	[85,86]
	Cytokeratin	↓ Downregulated	Reduced expression in ectopic epithelial cells leads to a loss of epithelial structure.	[81,87]
	Claudins (e.g., -3, -4, -7)	↓ Downregulated	Disruption of tight junction proteins in peritoneal lesions contributes to compromised epithelial integrity.	[20]
Mesenchymal Markers	N-cadherin	↑ Upregulated	Cadherin switches from E- to N-cadherin and promotes cell motility.	[81,87]
	Vimentin	↑ Upregulated	Associated with enhanced invasion and fibrosis.	[88,92]

**Table 2 ijms-26-07460-t002:** Performance of individual EMT-related biomarkers.

Biomarker	Relative Change	AUC (95%CI)	Sensitivity	Specificity	Reference
miR-17-5p	Decreased	0.74 (0.58–0.90)	NR	NR	[110]
miR-20a	Decreased	0.79 (0.65–0.93)	NR	NR	[110]
miR-22	Decreased	0.85 (0.71–0.98)	NR	NR	[110]
miR-200a	Decreased	0.75 (0.62–0.86)	90.6%	62.5%	[71,72]
miR-200b	Decreased	0.67 (0.53–0.79)	90.6%	45.8%	[71,72]
miR-141	Decreased	0.71 (0.57–0.82)	71.9%	70.8%	[71,72]
	Decreased	0.59 (0.51–0.67)	81.1%	37.5%	[111]
miR-199a	Increased	0.83 (0.73–0.92)	78.3%	76.0%	[112]
	Increased	0.62 (0.55–0.71)	54.6%	98.6%	[111]
	Increased	1.00 (no range)	91.4%	100%	[113]
miR-122	Increased	0.84 (0.75–0.92)	80.0%	76.0%	[112]
	Increased	0.72 (0.61–0.82)	45.8%	97.4%	[111]
	Increased	0.96 (no range)	95.6%	100%	[106]
miR-145	Decreased	0.88 (0.81–0.95)	70.0%	96.0%	[112]
	Decreased	0.76 (0.68–0.85)	85.3%	38.1%	[111]
miR-141	Decreased	0.85 (0.77–0.93)	71.7%	96.0%	[112]
miR-542-3p	Decreased	0.85 (0.77–0.94)	79.7%	92.0%	[112]
E-cadherin	Decreased	0.63 (0.33–0.81)	NR	NR	[107]
N-Cadherin	Increased	0.71 (0.45–0.86)	NR	NR	[107]
HIF-1α	Increased	0.74 (0.48–0.88)	NR	NR	[107]

AUC = area under the receiver operator curve; CI = confidence interval, NR = not reported.

**Table 3 ijms-26-07460-t003:** Performance of EMT-related biomarker panels.

Biomarker Panel	Relative Change	AUC (95%CI)	Sensitivity	Specificity	Reference
miR-200a miR-200b miR-141	Decreased Decreased Decreased	0.76 (0.63–0.87)	84.4%	66.7%	[72]
miR-199a miR-122 miR-145 miR-542-3p	Increased Increased Decreased Decreased	0.99 (0.98-1.00)	93.2%	96.0%	[112]
miR-141 miR-199a miR-122 miR-145 CA-125	Decreased Increased Increased Decreased Increased	0.94 (0.90–0.98)	81.8%	92.6%	[111]

AUC = area under the receiver operator curve; CI = confidence interval.

**Table 4 ijms-26-07460-t004:** Therapeutic candidates for targeting EMT in endometriosis.

Class	Molecule	Model	Findings	Reference
Isoflavonoids	Isoliquiritigenin 3,6-ihydroxyflavone	End1/E6E7 cells; mouse endometriosis model OMA stromal cells; SCID/rat models	↑ E-cadherin; ↓ N-cadherin, Snail, Slug; ↓ lesion size; ↑ BAX, caspase-3; ↑ E-cadherin, ↓ Twist, Slug; Notch pathway inhibition.	[121,122]
Plant Alkaloids	Tetramethylpyrazine	11Z cells, OMA stromal cells; mouse model	↑ E-cadherin; inhibited FMT; ↓ TGF-β1, α-SMA; ↓ p-Smad2/3; ↓ weight/fibrosis.	[123]
Terpenes	Parthenolide	Wistar albino rat model	↓ Lesion area; ↑ E-cadherin, ↓ vimentin; inhibited PI3K/AKT and β-catenin.	[124]
Polysaccharides	Fucoidan	End1/E6E7, Vk2/E6E7 cells; mouse model	↑ Proliferation/migration; ↑ E-cadherin; ↓ N-cadherin, Slug, Snail.	[92]
Biguanides	Metformin	OMA stromal cells; Wistar albino rat model	↓ β-catenin translocation; ↓ proliferation/invasion; ↑ apoptosis.	[125]
S1P Receptor Modulators	SEW2871, FTY720, JTE013	Mouse endometriosis model; Wistar rat model; Primary OMA stromal cells	↓ Lesion growth and fibrosis; ↓ collagen type I; ↑ E-cadherin; ↓ S1P1 signaling.	[126]

## Data Availability

Not applicable.

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
