# Peer review of "Insights into the Molecular Mechanisms and Signaling Pathways of Epithelial to Mesenchymal Transition (EMT) in the Pathophysiology of Endometriosis"

_ijms, 2025, doi:10.3390/ijms26157460_

Round 1
Reviewer 1 Report
Comments and Suggestions for Authors
The review article is very good. It is devoted to the study of the problem of epithelial-mesenchymal transition as a factor in the development of endometriosis. The review summarizes current knowledge about the role of epithelial-mesenchymal transition in the development of endometriosis, describes biomolecules for its diagnosis, and some pharmacological agents for therapeutic interventions in experimental cell models.
My comments:
1. I did not have enough material in Section 4.1. It does not mention obesity and molecules associated with the secretory activity of visceral adipocytes in abdominal obesity. There is also no mention of the role of microbiota and, in particular, Fusobacterium. There are articles on this problem in the world literature.
2. I did not have enough material in Section 7. This section is very poorly written. It needs to be expanded.
Author Response
Reviewer #1
The review article is very good. It is devoted to the study of the problem of epithelial-mesenchymal transition as a factor in the development of endometriosis. The review summarizes current knowledge about the role of epithelial-mesenchymal transition in the development of endometriosis, describes biomolecules for its diagnosis, and some pharmacological agents for therapeutic interventions in experimental cell models.
We are pleased with the positive comments.
Comment 1: I did not have enough material in Section 4.1. It does not mention obesity and molecules associated with the secretory activity of visceral adipocytes in abdominal obesity. There is also no mention of the role of microbiota and, in particular, Fusobacterium. There are articles on this problem in the world literature.
Response: We thank the reviewer for this insightful comment. We have revised Section 4.1 to include current evidence on the role of obesity and EMT-associated molecules, as well as the potential involvement of the microbiota, including Fusobacterium, in cancer and their possible relevance to EMT in endometriosis.
Comment 2: I did not have enough material in Section 7. This section is very poorly written. It needs to be expanded.
Response: We have expanded and revised Section 7 to include more additional information and to improve clarity.
Reviewer 2 Report
Comments and Suggestions for Authors
The manuscript entitled “ Insights into the Molecular Mechanisms and Signaling Pathways of Epithelial to Mesenchymal Transition (EMT) in the Pathophysiology of
Endometriosis” by Hosseinirad et al., is an interesting review that addresses de role of epithelial-to-mesenchymal transition (EMT) in endometriosis. Thus, the ectopic endometriosis lesions show traits of EMT, gaining migratory and invasive capabilities. EMT is driven by factors like hypoxia, TGF-β, and others, and its markers may offer new diagnostic and therapeutic options. This review is very well written, very well structured, easy to read and to follow, and it compiles extensive data on this subject.
I have a few comments to improve the text.
In general, I found that it is not always clear whether the information included refers specifically to endometriosis or is more general, not specifically related to endometriosis. Also, along the text, I found repeated information.
-Endometriosis description could be extended or expanded, deeper insights into the mechanism, growth factor, hormones, cytokines involved could be provided. It would be interesting to remark that the endometrial lesion has not only epithelial cells. It would be interesting to mention or comment on the fact that factors involved in EMT (TGFbeta, PDGF, hypoxia…) also affect (or not) other cells forming part of the endometroid lesion. It would be interesting to include information regarding the presence of these factors (TGFbeta, PDGF) in the location of the lesion.
In turn, line 100 et seq, could be eliminated, as the historic perspective is not necessary and
-it is not always clear whether the information included refers specifically to endometriosis or is more general, not specifically related to endometriosis.
-Line 160 et seq, It is not clear that TGFbeta has been shown to be upregulated in endometriosis. I wonder if mechanistic data have been obtained in endometrial cells showing the links: TGFbeta--- Snail--- EMT
Line 210. It is not clear the concept “estrogen….contributes….to hormone resistance”
Line 307 et seq. As progesterone resistance is a key feature of the endometriosis process, this point should be introduced earlier on in the manuscript
Section 4.3 could be lightened, as is somehow repetitive. E-cadherin and N-cadherin could be commented in parallel. Information regarding CK could be reduced and desmosomal paragraph could be eliminated. Why do authors refer to “activation” of vimentin… it could be more accurate “upregulation of vimentin”
Line 425. Confusing sentence, please revise
Lines 430-433, repetitive information
Line 433, is the SMM process found in endometroid lesions?
Line 436, Please, revise sentence. Wnt/beta-catenin is not a TGF-beta downstream pathway
It is not clear whether in endometriosis a fibrotic process can be found. The sentence “-“Ectopic lesions are more inflammatory and fibrotic that their counterparts …(line 423)” is quite vague
Section 7. Please include some information about how the diagnosis of endometriosis is done at present time, and include a critical reflection on whether it would be possible to determine microRNA in the clinical practice.
Section 8. Please include some information about the treatment of endometriosis is done at the present time. Authors must remark that molecules that they commented on are based in preclinical evidence
Author Response
Reviewer #2
The manuscript entitled “ Insights into the Molecular Mechanisms and Signaling Pathways of Epithelial to Mesenchymal Transition (EMT) in the Pathophysiology of Endometriosis” by Hosseinirad et al., is an interesting review that addresses de role of epithelial-to-mesenchymal transition (EMT) in endometriosis. Thus, the ectopic endometriosis lesions show traits of EMT, gaining migratory and invasive capabilities. EMT is driven by factors like hypoxia, TGF-β, and others, and its markers may offer new diagnostic and therapeutic options. This review is very well written, very well structured, easy to read and to follow, and it compiles extensive data on this subject. I have a few comments to improve the text. In general, I found that it is not always clear whether the information included refers specifically to endometriosis or is more general, not specifically related to endometriosis. Also, along the text, I found repeated information.
We are grateful for the encouraging feedback. In response to the reviewer’s comments, we have revised the manuscript to enhance its clarity and quality.
Comment 1: Endometriosis description could be extended or expanded, deeper insights into the mechanism, growth factor, hormones, cytokines involved could be provided. It would be interesting to remark that the endometrial lesion has not only epithelial cells. It would be interesting to mention or comment on the fact that factors involved in EMT (TGFbeta, PDGF, hypoxia…) also affect (or not) other cells forming part of the endometroid lesion. It would be interesting to include information regarding the presence of these factors (TGFbeta, PDGF) in the location of the lesion.
Response: We have expanded the introduction to include detailed roles of growth factors (TGFβ, PDGF), hormones, and cytokines in endometriosis and EMT. We clarified that endometriotic lesions are composed of several distinct cell types that reflect their complex, heterogeneous nature. These include epithelial, stromal, endothelial, nerve, and immune cells.
Comment 2: In turn, line 100 et seq, could be eliminated, as the historic perspective is not necessary and -it is not always clear whether the information included refers specifically to endometriosis or is more general, not specifically related to endometriosis.
Response: We revised the text to clarify when the discussion is based on endometriosis-specific data versus general EMT knowledge.
Comment 3: Line 160 et seq, It is not clear that TGFbeta has been shown to be upregulated in endometriosis. I wonder if mechanistic data have been obtained in endometrial cells showing the links: TGFbeta--- Snail--- EMT
Response: We clarified in Section 4.1.1 that TGFβ1 is upregulated in endometriotic lesions and added references demonstrating that TGFβ induces Snail expression and promotes EMT in endometrial epithelial and stromal cells in vitro.
Comment 4: Line 210. It is not clear the concept “estrogen….contributes….to hormone resistance”
Response: The sentence has been revised to explain that local estrogen production can disrupt progesterone receptor signaling, contributing to progesterone resistance in endometriosis.
Comment 5: Line 307 et seq. As progesterone resistance is a key feature of the endometriosis process, this point should be introduced earlier on in the manuscript
Response: We have moved the discussion of progesterone resistance to Section 4.1.2.
Comment 6. Section 4.3 could be lightened, as is somehow repetitive. E-cadherin and N-cadherin could be commented in parallel. Information regarding CK could be reduced and desmosomal paragraph could be eliminated. Why do authors refer to “activation” of vimentin… it could be more accurate “upregulation of vimentin”
Response: Section 4.3 has been shortened. Cadherins are now discussed in parallel, details on CK have been reduced, the paragraph on desmosomes has been removed, and the phrase “activation of vimentin” has been corrected to “upregulation of vimentin.”
Comment 7: Line 425. Confusing sentence, please revise
Response: The sentence at line 425 has been revised for clarity to better explain the roles of inflammation and fibrosis in the development and progression of ectopic lesions.
Comment 8: Lines 430-433, repetitive information
Response: We have removed the repetitive information previously presented in lines 430–433 to improve clarity and reduce redundancy.
Comment 9: Line 433, is the SMM process found in endometroid lesions?
Response: We clarified that smooth muscle metaplasia (SMM) is observed in specific lesion types, particularly in rectovaginal endometriosis.
Comment 10:Line 436, Please, revise sentence. Wnt/beta-catenin is not a TGF-beta downstream pathway. It is not clear whether in endometriosis a fibrotic process can be found. The sentence “-“Ectopic lesions are more inflammatory and fibrotic that their counterparts …(line 423)” is quite vague.
Response: We corrected the sentence to clarify that Wnt/β-catenin is not a downstream pathway of TGFβ and revised the fibrosis-related statement to be more specific and supported by evidence.
Comment 11: Section 7. Please include some information about how the diagnosis of endometriosis is done at present time, and include a critical reflection on whether it would be possible to determine microRNA in the clinical practice.
Response: We added a section discussing current diagnostic methods—including clinical assessment, imaging, and laparoscopy—and addressed the potential and limitations of using microRNAs in clinical practice.
Comment 11: Section 8. Please include some information about the treatment of endometriosis is done at the present time. Authors must remark that molecules that they commented on are based in preclinical evidence
Response: We added a brief overview of current treatment options and clarified that most of the molecular targets discussed in the review are supported by preclinical data and have not yet been translated into clinical applications.
Reviewer 3 Report
Comments and Suggestions for Authors
Dear Authors, I appreciate the work and probably the team work you have made for this review.
There are points to appreciate and some points to correct, maybe, for a better reading and understanding of endometriosis.
I also like the specificity of your approach, there is lots of information, aspects to discuss, mechanisms and hypothesis to elaborate.
Nevertheless, there are some aspects that could be improved, on my opinion.
I suggest to reorganise, summarise and highlight the important data from section 4. I suggest to rename it, for Manifestations of EMT is very general and at the same time all the subsections of section 4 are very precise, example 4.1.
Rows 270-277 belong to conclusions, if not maybe reformulate the beginning of the phrase.
Also, rename and reorganise section 5, 6; for section 7, I suggest simply Biomarkers.
Section 9 should be divided in Future DIrections and separately, Conclusions. And, Conclusions should be very sharp, suggestive, and resumed.
My conclusion , after carefully reading your work, is that you need a better re organisation and a much clear vue of all big aspects, and when needed, of all sub aspects that you need to discuss.
Good luck!
Comments on the Quality of English Language
No comment.
Author Response
Reviewer #3
Dear Authors, I appreciate the work and probably the team work you have made for this review. There are points to appreciate and some points to correct, maybe, for a better reading and understanding of endometriosis. I also like the specificity of your approach, there is lots of information, aspects to discuss, mechanisms and hypothesis to elaborate.
Nevertheless, there are some aspects that could be improved, on my opinion.
We are pleased with the positive comments. We have incorporated the reviewer’s suggestions into this revised manuscript, which has significantly improved the clarity and overall quality.
Comment 1: I suggest to reorganise, summarise and highlight the important data from section 4. I suggest to rename it, for Manifestations of EMT is very general and at the same time all the subsections of section 4 are very precise, example 4.1.
Response: Taking the reviewer’s suggestions, we have summarized and highlighted the important data at the beginning of Section 4, reorganized the section into several clearer subsections, and renamed it to better reflect its content. Additionally, Figure 1 and Table 1 have been summarized and integrated within Section 4 to enhance clarity and cohesion.
Comment 2: Rows 270–277 belong to conclusions; if not, maybe reformulate the beginning of the phrase.
Response: We have revised and reformulated the beginning of this section to improve clarity.
Comment 3: Also, rename and reorganize Sections 5 and 6; for Section 7, I suggest simply “Biomarkers.”
Response: We have renamed and reorganized Sections 5 and 6 for improved clarity and flow. Section 7 has also been retitled as “Biomarkers” as recommended.
Comment 4: Section 9 should be divided into “Future Directions” and “Conclusions” separately. Conclusions should be very sharp, suggestive, and concise.
Response: We have divided Section 9 into two separate sections: “Future Directions” and “Conclusions.”. The Conclusions section has been revised to be clearer, more focused, and concise.
Comment 5: My conclusion, after carefully reading your work, is that you need better reorganization and a much clearer view of all big aspects and, when needed, of all sub-aspects that you need to discuss.
Response: We appreciate this overall feedback. We have extensively reorganized the manuscript to improve clarity, logical flow, and coherence, ensuring that both the broader aspects and specific subtopics are clearly articulated and thoroughly discussed.